# Multi-Season Genome-Wide Association Study Reveals Loci and Candidate Genes for Fruit Quality and Maturity Traits in Peach

**DOI:** 10.3390/plants15020189

**Published:** 2026-01-07

**Authors:** María Osorio, Arnau Fiol, Paulina Ballesta, Sebastián Ahumada, Pilar Marambio, Pamela Martínez-Carrasco, Rodrigo Infante, Igor Pacheco

**Affiliations:** 1Laboratorio de Biología Vegetal e Innovación en Sistemas Agroalimentarios—Grupo Nutribreeding, Instituto de Nutrición y Tecnología de Alimentos Dr. Fernando Monckeberg Barros (INTA), Universidad de Chile, El Líbano #5524, Macul 7830490, Chile; maria.osorio@inta.uchile.cl (M.O.); pballesta@udla.cl (P.B.);; 2Núcleo de Investigación en Sustentabilidad Agroambiental (NISUA), Facultad de Medicina Veterinaria y Agronomía, Universidad de las Américas, Manuel Montt #948, Providencia 7500975, Chile; 3Departamento de Producción Agrícola, Facultad de Ciencias Agronómicas, Universidad de Chile, Santa Rosa #11315, La Pintana 8820808, Chile

**Keywords:** *Prunus persica* [L.] Batsch, peach, genome-wide association study (GWAS), fruit maturity date, chlorophyll index (IAD), soluble solids content (SSC), fruit weight (FW), linkage disequilibrium (LD), SNP markers, candidate genes

## Abstract

Peaches are a fruit crop with global importance due to their economic value. Fruit quality (e.g., weight, soluble solids content (SSC)) and phenology traits (e.g., maturity date) are essential for generating novel varieties. Nevertheless, modern germplasm’s narrow genetic diversity hampers breeding efforts to enhance these traits. To identify genetic markers helpful for marker-assisted breeding, this work leveraged a diverse panel of 140 peach commercial cultivars and advanced breeding lines phenotyped across three harvest seasons for the maturity date (MD), chlorophyll absorbance (IAD), SSC, and fruit weight (FW). Genotypic data were generated via ddRADseq, identifying 5861 SNPs. A rapid linkage disequilibrium decay (critical r^2^ = 0.308 at 950 kb) was determined, and a population structure analysis revealed two admixed genetic clusters, with phenotypic distributions influenced by seasonal environmental factors. A total of 599 marker–trait associations were detected by using single and multi-year analysis, and for each trait the surrounding genomic regions explored to identify potential candidate genes annotated with functions related to the trait under study, and expressed in peach fruits. This study highlights multiple *loci* potentially responsible for phenotypic variations in plant phenology and fruit quality, and provides molecular markers to assist peach breeding for fruit quality.

## 1. Introduction

The Peach *Prunus persica* (L.) Batsch is a deciduous tree belonging to the subgenus *Amygdalus* within the Rosaceae family and serves as the model species for the *Prunus* genus [1]. The species originated in China, as evidenced by archaeological findings dating back to approximately 6000 BC [2]. The species has demonstrated a remarkable capacity for climate adaptation and maintains a high productivity in a broad range of agricultural zones [3]. Historically, the peach crop has been the most significant and widely cultivated species within the genus *Prunus*. Chile is the leading exporter of peaches in the Southern Hemisphere [4], underscoring the economic importance of this crop in global trade.

Domestication and breeding have played a crucial role in shaping the genetic diversity of peaches through the selective breeding of desirable traits. Over centuries, breeders have prioritized attributes such as fruit size, color, pulp firmness, harvest timing, and postharvest quality to meet market demands and optimize commercialization [5,6,7]. In addition, peach trees are predominantly self-pollinated [8], which has significantly reduced the crop’s genetic diversity and variability [9,10,11]. Accordingly, previous studies have shown that peaches exhibit low levels of polymorphism compared to other crops from the same genus [12,13,14,15].

Given its economic importance and increasing global demand, peach production must overcome several challenges to produce high-quality, high-volume fruit. These include, among others, abiotic and biotic stress factors, the need to differentiate products in competitive markets, ensuring sustainable production systems, and optimizing fruit quality, specifically in postharvest [16,17]. To address these issues, breeding programs aim to develop new varieties with greater stress tolerance, extended shelf life, and adaptability to diverse agroclimatic conditions [18,19]. For this reason, breeding programs continually produce new varieties that meet the community’s demands. Given the long generation time of the peach, marker-assisted selection has been proposed as a viable alternative for targeted selection of some traits, based on allelic information from markers associated with the trait of interest. Genomic-Wide Association Studies (GWAS) have been proven to be a powerful approach to discover such markers from genotypic and phenotypic information of a population: for example, in apple [20], identified through GWAS the NAC18.1 D5Y SNP as a significant determinant of harvest date and firmness, which is now used to assist selection for firmer and slow-softening cultivars; in cherry [21] combined GWAS with other genetic approaches to identify SNP haplotypes strongly associated with fruit cracking susceptibility.

One of the most critical aspects of peach production is determining the optimal stage of fruit maturity, as an improper harvest time can significantly impact fruit quality. Accurate maturity assessments are crucial for ensuring efficient harvest planning and meeting market quality standards. Currently, both destructive and non-destructive methods are used to estimate maturity, including firmness measurements, chlorophyll content analysis, and spectroscopic techniques [22,23,24,25]. Implementing these techniques not only enhances the quality of the fruit delivered to consumers but also supports breeding strategies aimed at improving uniform ripening and postharvest performance. One of the instruments used in estimating the optimal harvest date is the Sinteleia DA meter, which measures wavelength absorption at 670 and 720 nm and returns the index of absorbance difference (IAD). This value reflects the content of chlorophyll a in the fruit, whose degradation is directly related to the advance of maturation [26]. This technique is reported to help establish the optimal harvest time in peaches [25,27,28,29]. However, because there is no specific value that investigators recommend for the species, it is not a reliable indicator when used as the sole measure of maturity. Previous studies have provided recommendations for each type of peach, such as melting or non-melting peaches; similarly, in apples [30], the optimal IAD maturity values are cultivar-dependent.

This study aimed to detect genetic associations and candidate genes controlling fruit quality and phenological traits in a diverse panel of 140 peach trees. This panel comprises commercial cultivars and advanced breeding lines from the stone fruit breeding program at the University of Chile, evaluated over three harvest seasons for fruit weight (FW), soluble solid content (SSC), maturity date (MD), and chlorophyll content (IAD). The identification of genetic markers and candidate genes associated with key phenological and fruit quality-related traits in peach across several seasons provides a valuable genetic background for future marker-assisted selection in breeding programs.

## 2. Results

### 2.1. Genotyping Results

The sequencing generated a mean of 4.3 million reads per sample, which allowed to identify 5861 high-quality single-nucleotide polymorphisms (SNPs) markers. The mean read depth and genotyping rate per sample on those variant positions was 52.54 and 0.97, respectively. A summary of the obtained ddRAD sequencing raw data quality for the 140 analyzed accessions is shown in Appendix A. The mean number of SNPs per chromosome was 732.63, with the highest SNP count observed on chromosome 1 (*n* = 956 SNPs) and the lowest on chromosome 5 (*n* = 452 SNPs) (Table 1). The SNPs were distributed across the eight chromosomes of *P. persica* with a mean density ranging from 2.00 (Chr1) to 3.32 (Chr4) markers every 100 kilobases (Figure 1).

### 2.2. Linkage Disequilibrium

The mean linkage disequilibrium value for all eight chromosomes was *r*^2^ = 0.1475 (Table 1), with the highest value on Chr5 (*r*^2^ = 0.257) and the lowest value on Chr7 (*r*^2^ = 0.116). The whole-genome Linkage Disequilibrium (LD) decay pattern is shown in Figure 1. The calculated LD critical value was *r*^2^ = 0.30 (*p*-value < 0.01), corresponding to linkage equilibrium at a mean distance of 950 kb according to the method reported by Breseghello and Sorells [31].

### 2.3. Phenotypic Characterization of the Peach Germplasm

The distribution for each trait and season is shown in Figure 2. The fruit weight (FW) exhibited a unimodal distribution across all years under consideration, with means of 146.53, 168.55, and 191.81 for seasons 2020–2021, 2021–2022, and 2023–2024, respectively. The minimum and maximum values for the seasons 2020–2021, 2021–2022, and 2023–2024 were 42 g and 387 g, 33.1 g and 405.7 g, and 10.7 g and 383.16, respectively. The soluble solid content (SSC) distribution for seasons 2020–2021 and 2023–2024 was similar, with a mean value of 15.90 °Brix and 16.05 °Brix, respectively, and a range of values from 4.4 °Brix to 29.8 °Brix and from 5.9 °Brix to 32.3 °Brix. In particular, the SSC for the 2020–2021 season displayed a bimodal distribution with peaks at 13.8° and 18.5° Brix, and a trimodal distribution for the 2021–2022 season (three peaks at 9.7°, 14.6° and 20.5° Brix). According to the visual representation, the 2023–2024 season exhibited a bimodal distribution with two peaks at 14° and 19.8° Brix. In the case of maturity date (MD), the 2020–2021 season exhibited a bimodal distribution, with peaks at 156 and 207 Julian days (JDs), and mean, minimum, and maximum values of 192.029 JDs, 153 JDs, and 259 JDs, respectively. The 2021–2022 season showed a multimodal (four peaks) pattern at 167, 186, 213, and 235 JDs, with similar frequencies of 0.20, 0.39, 0.20, and 0.19, respectively; and an overall mean value of 198.407 JDs (range: 161 to 259 JDs). Consistently, the 2023–2024 season exhibited a similar multimodal distribution with peaks at 167, 182, 202, and 238 JDs, and a mean value of 200.066 JDs (range: 152 to 299 JDs). The IAD evaluated in the 2020–2021 and 2021–2022 seasons displayed a unimodal distribution, with means of 0.718 and 0.836, respectively, and ranges of 0.01 to 3.01 and 0.01 to 2.25, respectively. For the 2023–2024 season, the IAD exhibited a multimodal distribution with peaks at 0.33, 0.58, and 1.2, and a mean of 0.911 (range: 0.02–2.31).

All traits were significantly and positively correlated (*p* < 0.001) between seasons (Appendix A). FW assessed in the 2020–2021 season was highly correlated with those evaluated in the 2021–2022 and 2023–2024 seasons (r = 0.58 and 0.64, respectively). Fruit Weight (FW) assessed in 2021–2022 was moderately correlated with that evaluated in 2023–2024 (r = 0.47). SSC assessments between seasons were moderately correlated, with correlation values ranging from 0.38 (between 2020–2021 and 2023–2024 seasons) to 0.57 (between 2020–2021 and 2021–2022 seasons). The MD was the trait with the highest correlation value between seasons, where correlation values ranged from 0.85 (between 2020–2021 and 2023–2024) to 0.96 (between 2021–2022 and 2023–2024), while IAD fluctuated between 0.55 (between seasons 2020–2021 and 2023–2024) to 0.62 (between 2021–2022 and 2023–2024). FW was positively and moderately correlated (*p* < 0.05) with MD for all seasons (r = from 0.23 to 0.39), except for FW for the 2021–2022 season and MD assessed in the 2023–2024 season. No significant correlations were detected between FW and IAD or SSC, with the exception that FW for the 2021–2022 season was significant (*p* < 0.01) and negatively correlated with IAD for the 2020–2021 season (r = −0.25). MD for the 2020–2021 season was moderately and positively correlated (*p* < 0.05) with all IADs assessed across the three seasons (r = 0.2–0.39), while MD for the 2021–2022 and 2023–2024 seasons was only positively correlated (*p* < 0.001) with IADs for the 2020–2021 (r = 0.35 and 0.36, respectively) and 2021–2022 seasons (r = 0.33 and 0.33, respectively). In general, SSC was not significantly correlated with any of the traits evaluated, except for SSC for the 2020–2021 season with MD for the 2020–2021 seasons (r = −0.21), 2023–2024 (r = −0.27), and SSC for the 2023–2024 season with MD evaluated in the 2021–2022 season (r = 0.19). The calculated broad-sense heritability (H^2^) for each trait was 0.98 for the MD, 0.93 for the FW, 0.87 for the SSC, and 0.81 for the IAD.

### 2.4. Population Structure

According to Evanno’s method, the population was found to be composed of two genetic clusters (K = 2). A total of 115 individuals (82.14% of the population) were assigned to either one of the two groups (Q1 or Q2) with a membership probability (Q) higher than 75%: 92 individuals (65.71%) were assigned to Q1, 23 (16.43%) to Q2, and 25 (17.86%) corresponded to admixed individuals (Figure 3a).

The first five components of the genetic principal component analysis (PCA) explained 44.6% of the variation, with the first component responsible for 15.01% and the second accounting for 11.06% of the total variation. The first principal component was enough to differentiate the two genetic clusters defined using the Evanno method (Figure 3b), with the admixed individuals clustering between the two differentiated groups. The second principal component further differentiated five individuals within Q2, which clustered together and were separated from the majority of Q2 individuals.

The Maximum Likelihood (ML) phylogenetic tree elucidated the evolutionary relationships among the 140 individuals (Figure 4). The five individuals that separated from the remaining Q2 individuals in the second component of the PCA were grouped inside a cluster that contained most (20 out of 23) individuals of this genetic cluster. The three remaining Q2 individuals formed a closely grouped clade that also included 13 admixture and 16 Q1 individuals. The rest of the phylogenetic tree comprises the remaining Q1 individuals, along with admixture individuals that tend to form smaller clusters due to their similar mixed backgrounds.

### 2.5. Marker–Traits Associations and Discovery of Candidate Genes

In this study, we identified a total of 599 marker–trait associations for all studied traits and seasons (Appendix A; Figure 5). For the FW evaluated in the 2020–2021, 2022–2023, and 2023–2024 seasons, 15, 28, and 35 marker–trait associations (MTAs) were detected, respectively, which explained between 7% and 15%, 6% and 12%, and 9% and 19% of the phenotypic variation (PVE). Most of the MTAs were located on chromosomes 8, 6 and 6 for the 2020–2021, 2022–2023 and 2023–2024 seasons, respectively. In the case of the multi-season analysis, 79 MTAs were detected, which explained between 7 and 15% of the phenotypic variation in FW and were mainly located on chromosome 6 (Figure 5), between SNPs Pp06_2864162 and Pp06_4943804, where the strongest signals were observed in positions Pp06_2988972, Pp06_3223902 and Pp06_3642221 (Table 2). Based on the *P. persica* reference genome, 19 of the SNPs significantly associated with FW were situated near genes potentially related to cell cycle signaling and regulation, cell elongation and growth, and cell wall structure (i.e., biosynthesis, modification, remodeling; Table 2). For instance, the three MTAs detected on chromosome 6 (SNPs Pp06_2988972, Pp06_3223902, and Pp06_364222) lie within the same region as a gene expressed at stage S3 [32], which encodes an endo-1,3-β-glucosidase 9 (*Prupe.6G050800*) involved in cell wall remodeling. Other MTAs with minor significance signals are in the same LD region of genes such as OVATE domain-containing protein (3 SNPs), Auxin-binding protein T85 (4 SNPs), Xyloglucan endotransglucosylase/hydrolase (7 SNPs), and Pectin methylesterase *CGR3 *(1 SNP) (See Table 2).

SSC was the trait with the lowest number of detected MTAs (92 SNPs). Twenty-four, twelve and eight MTAs were identified for the seasons 2020–2021, 2021–2022 and 2023–2024, respectively. Particularly, the MTAs for the 2020–2021, 2021–2022, and 2022–2023 seasons explained between 6% and 18%, 6% and 10%, and 8% and 10% of the phenotypic variation in SSC, respectively. Based on the multi-season GWAS, 48 MTAs were detected, located on chromosomes 1 (1 MTA), 4 (31 MTAs), 5 (11 MTAs), and 7 (5 MTAs). The most significant MTA for this trait was located at SNP Pp05_16778038 (PVE = 15.9%), near a gene encoding an *AGAMOUS-like* MADS-box protein *AGL8* that is expressed in peach fruit. Moreover, one SSC-associated MTA (Pp01_12736773) was located near a Sucrose-phosphate synthase gene (*Prupe.1G159700*) and one MTA (Pp04_16076936) near a gene coding for a Sorbitol dehydrogenase (*Prupe.4G240300*). All these MTAs were detected both in the 2020–2021 season and multi-year analyses (see Appendix A).

For the MD evaluated in the 2020–2021, 2022–2023, and 2023–2024 seasons, 74, 60, and 53 MTAs were detected, respectively, with PVEs between 7% and 20%, 7% and 19%, and 8% to 15%. The significantly associated SNPs were located on chromosomes 1, 2, 3, 4, 6, 7 and 8, and only five MTAs for the 2023–2024 season were located on chromosome 5. In the case of the multi-season analysis, 46 MTAs were detected, explaining from 6 to 20% of the phenotypic variation in MD. Through LD-based region scanning, several MTAs were detected located near candidate genes. The strongest MTA observed in this study (*p*-value = 2.53 × 10^−6^) was located in SNP Pp04_11112825 in 2020–2021, 2021–2022, and the multiyear dataset (Appendix A). The 950 bp region around this SNP contains a *NAC* transcription factor involved in peach senescence. Other MTAs contain genes involved in pectin catabolic process, fruit ripening, and response to ethylene, among others (Table 2).

SNP-based associations accounted for 7% to 19% of the PVE of IAD evaluated in the 2020–2021, 2021–2022, and 2023–2024 seasons. The 35 MTAs for the 2020–2021 season were mapped in all the *Prunus* chromosomes, except on chromosome 7. Twenty-nine SNPs were significantly associated with IAD for the 2021–2022 season, of which about 48% of the MTAs were located on chromosome 2. For the 2023–2024 season, about 58% of the total MTAs were located on chromosome 6. The multi-season GWAS revealed 29 MTAs, accounting for 7–19% of the phenotypic variation in the trait, with ~38% (11 SNPs) of the total MTAs located on chromosome 6. The strongest MTA effect was detected in the chromosome 2 (SNP Pp02_19688605; *p* = 6.83 × 10^−6^; 18.8%) in the multiyear dataset and, although with minor significance, in the 2020–2021, 2021–2022 and 2023–2024 seasons (*p* = 7.66 × 10^−5^, 7.61 × 10^−4^ and *p* = 2.37 × 10^−4^, respectively); this MTA contains a “Plastid-lipid associated protein PAP/fibrillin family protein” coding gene, responsible for chloroplast to chromoplast remodeling. Three MTAs, based on the multi-season dataset, were found in chromosome 1 (Pp01_3330340) close to a pheophytinase gene (*Prupe.1G353100*) and two of them in chromosome 6 (Pp06_857629 and Pp06_858783) near a pheophorbide a oxygenase gene (*Prupe.6G113600*). Also, three SNP markers (Pp08_583586, Pp06_11768361 and Pp06_11906644) in the single-season analysis were distributed on chromosomes 6 and 8 and located near genes coding for proteins with putative chlorophyllase activity (*Prupe.6G143300*) and regulation of chlorophyll degradation (*Prupe.8G010500*) in the single-season analysis (Table 2).

Considering the multi-season analysis, 20 SNP markers were significantly associated with more than one trait (Appendix A), which were located exclusively on chromosomes 4 and 6. Specifically, 12 markers were significantly associated with FW and MD, two markers were significantly associated with FW and IAD, and six markers were related to both FW and IAD, as well as MD.

## 3. Discussion

The global demand for new and improved peach varieties is increasing due to consumers’ new needs and the challenges posed by climate change. To accelerate the generation of elite cultivars, it is necessary to unravel the genetic mechanisms of important agronomic traits and unlock molecular breeding strategies [33]. In this context, here the fruit weight (FW), solid soluble content (SSC), maturity date (MD) and chlorophyll absorbance index (IAD) have been evaluated for three consecutive seasons in 140 peach trees from the stone fruit breeding program of the University of Chile, aimed at detecting marker–trait associations (MTAs) for these relevant traits. The study’s salient points and their implications for peach breeding and genetics is discussed below.

### 3.1. Genotyping and Linkage Disequilibrium

In this study, a total of 5861 high-quality single-nucleotide polymorphism (SNP) markers were detected by generating ddRAD sequencing reads. This genotyping platform has already proven effective for genetic characterization, genome-wide association, and genomic selection in *Prunus* crop collections, including peach and Japanese plum [34,35,36]. One of the common drawbacks of ddRAD is that a high number of *loci* contain missing data [37], which was mitigated here by applying restrictive filters to keep only the best high-quality SNPs. Because the library construction relies on two selected restriction enzymes, the genotyped *loci* are randomly generated. Accordingly, the selected SNPs were well distributed across the eight chromosomes of peach, with a marker count per chromosome proportional to their sizes.

To estimate the linkage disequilibrium (LD) in the population under study is crucial for understanding its genetic structure and the feasibility of performing genome-wide association studies (GWAS). In the studied population, the mean *r*^2^ value of 0.147 indicated a moderate level of LD. However, noticeable variations were detected among chromosomes, with Chr5 showing the highest *r*^2^ value (0.257) and Chr7 the lowest (0.116). These differences can be attributed to factors like variable recombination rates across the genome, selection history, population structure, past introgression events and bottlenecks [11,38,39].

It has been observed that a significant number of studies conducted on peach trees have demonstrated a rapid decay of LD [40,41,42]. However, in this instance, the observed behavior does not correspond to the findings of these studies. The underlying reasons for this discrepancy are likely due to the inherent characteristics of the samples and the magnitude of the population size. For example, in Osorio et al. (2025) [35], the LD of a Japanese plum population was compared to a core collection (~10% the original population size) originated from it, and revealed discrepancy in the patterns of LD and the critical value, with the reduced group demonstrating a higher critical *r*^2^ value even though the samples originated from the same population.

Upon analysis of the LD decay, it is evident that there is a rapid decline in the LD values, which reach a critical value of *r*^2^ = 0.308 at a physical distance of 950 kilobases. This rapid LD decay may be advantageous for GWAS, as it allows for more precise localization of candidate genes. However, it simultaneously underscores the need for a high genetic marker density to capture relevant variation and ensure that markers are sufficiently close to causal *loci*. The presence of this LD decay pattern provides a solid basis for subsequent genome-wide association analyses, allowing the delineation of search windows for candidate gene identification, which is implemented here with a 475 kb window upstream and downstream of significant markers.

### 3.2. Phenotypic Variability

In general, the phenotypic distribution of most traits was observed to be variable among evaluation seasons. The distribution of FW was similar between seasons, which suggests that environmental effects have a minimal influence on this trait. In contrast, in an F1 peach population from 9 crosses, it was determined that FW has a narrow-sense heritability from low to moderate [43]. On the other hand, genomic prediction studies in peach suggest that the heritability of FW can fluctuate between 0.22 and 0.78, depending on the evaluated crossing scheme [44]. In fact, according to da Silva Linge et al. (2015) [45], the FW can be a trait subject to transgressive segregation, which leads to highly variable trait inheritance depending on the effect of the parental lines.

SSC is a complex trait characterized by low inheritance, as its expression is strongly influenced by environmental factors, such as canopy position, water availability, light radiation, plant nutrition, and others [33]. Narrow and broad sense heritability estimates indicated that the SSC may have relatively low genetic control in contrast to other fruit quality-related traits in peach [43], which is consistent with the results of the present study. Indeed, SSC exhibited more complex distribution patterns, displaying a bimodal distribution in 2020–2021 and 2023–2024, with different peaks, while the 2021–2022 season presented a trimodal distribution.

Along with fruit quality traits, several other features of peaches, such as reduced chilling requirements and expanded environmental ranges, are important for plant breeding [7]. Specifically, the fruit development period and ripening dates are key for crop scheduling. In this study, it was found that the phenotypic distribution of MD measured in the 2020–2021 season was notably different from other seasons. Notably, the 2021–2022 and 2023–2024 seasons showed similar phenotypic means regarding MD, with mode values differing by no more than ±10 days. In *Prunus*, MD is regarded as a highly heritable trait [46,47,48], which aligns with the similar pattern observed between the 2021–2022 and 2023–2024 seasons; however, some studies suggest it is not necessarily controlled by additive effects in peach [43], and its expression is also influenced by environmental factors [49,50].

The present study examined the variability of IAD, an indicator of chlorophyll content in the fruit, whose content dramatically decreases as fruit senescence begins, thus associating it with fruit maturity. Since this measure does not have a standard value across varieties, we searched for genetic determinants related to chlorophyll degradation. The IAD initially exhibited a unimodal distribution in the first two seasons but transitioned to a multimodal distribution with three peaks in the last season. This change suggests that, similarly to SSC, environmental factors may influence trait values, potentially because of variations in environmental cues that impact chlorophyll degradation pathways. Although peach fruit color has been described to be relatively genetically controlled [43] and is typically stable and cultivar-specific, several studies suggest that environmental factors and external treatments can modulate fruit peel color and increase chlorophyll degradation. A detailed review is provided in Shin et al. (2023) [51].

Although some traits exhibited differential distribution between seasons, phenotypic correlation analysis revealed that all traits were significantly and positively correlated between seasons, demonstrating that the cultivars and selections maintained in the study trial probably did not exhibit a strong genotype-environment interaction. In agreement, the calculated H^2^ values ranged between 0.98 (MD) and 0.81 (IAD), indicating that most of the phenotypic variability in the studied population is due to the genotypic variability of its individuals. Although those values are specific to the population, similar results were calculated in Ksouri et al. (2025) [52], coinciding also in the MD also being the trait with the highest heritability, and the FW with higher value than the SSC. Regarding the relationship between traits, FW was found to be positively correlated with MD in all seasons, a finding also reported by Dirlewanger et al. (1999) [53], Rawandoozi et al. (2021) [43], and Eduardo et al. (2011) [54]. In fact, Eduardo et al. (2011) determined genetic regions associated with MD that were pleiotropic for FW [54]. This study reported a significant positive correlation between MD and IAD, suggesting that in the considered genetic background, maturity date is directly proportional to chlorophyll content. According to Pinto et al. (2015) [28], peach fruits that ripen later maintain higher IAD values for longer, in contrast to earlier fruits. According to the phenotypic distribution of MD for the 2021–2022 and 2023–2024 seasons, where the correlation between IAD and MD is most significant, most trees tend to have later MD.

### 3.3. Population Structure

Consistent with previous studies on *P. persica*, the population structure revealed two genetically homogeneous groups (K = 2). This finding was previously reported by Jiang et al. (2022) [55] in a comparison of 16 Chinese populations of *P. persica*, which exhibited low within-population genetic variation and a high level of genetic differentiation between the two clusters. In a germplasm comprising 94 genotypes, including local Spanish and modern cultivars, Font i Forcada et al. (2019) [56] determined a population structure consisting of two genetically homogeneous groups using ~8K SNP markers.

Although both Evanno-based structure analysis [57] and PCA supported the presence of two homogeneous genetic groups, a notable inconsistency was identified for five individuals assigned to cluster Q2 that were isolated from the Q2 core according to the PC2. The differences can be explained by the fact that both methods, in addition to having different analytical assumptions (i.e., Bayesian and frequentist approaches), the Bayesian method relies on capturing ancestral proportions of individuals’ alleles in a general way. At the same time, PCA can reveal recent divergences or unique allelic combinations that differentiate individuals. Consequently, the ML dendrogram classified the five individuals into a common group, primarily close to the other individuals assigned to group Q2, which is consistent with the premise that the ML method defines clusters by recognizing genetic profiles and selecting the tree with the highest probability. Discrepancies between clustering methods have been previously reported in other studies in *P. persica*. Cirilli et al. (2017) [58] reported that a panel of commercial cultivars was structured into two clusters, with some genotypes not definitively assigned to these two groups according to PCA. A similar finding was reported by Vodiasova et al. (2025) [42] for a collection of 161 cultivars and hybrids, where seven subgroups were identified in the population, and PCA failed to assign individuals to each group correctly. Given discrepancies across previous studies, this study explicitly accounted for structural effects in association analyses to reduce false positives in QTL detection. In particular, population genetic structure, along with relatedness, is a relevant factor in peach because, despite the species’ limited genetic diversity, its populations exhibit high levels of structuring and differentiation [40,54].

### 3.4. Association Analysis

In the Mixed Linear Model (MLM) analysis, a total of 2 markers were detected with significance values below the very stringent Bonferroni genome-wide significance threshold (*p*-value < 8.5 × 10^−6^), none of which were detected in the single-season analysis. This is a relatively common challenge in GWAS for complex traits in perennial crops and in relatively small populations [59,60,61], including the present study with moderate sample size, and is attributed to different factors. Firstly, the polygenic nature of the traits, in which many genes with minor effects contribute to the phenotype, and secondly, the strong influence of Genotype-by-Environment (GxE), as already demonstrated in peach for the SSC trait [62]. Consequently, potential genetic signals are obscured by insufficient statistical power. This was evidenced here by the fact that some genetic signals were not detected in all three considered seasons in the single-year analysis due to having a low effect and/or the environmental effect. The multi-year approach using the Best Linear Unbiased Predictors (BLUEs) values mitigates the GxE effects and provides a more stable estimate of the genetic potential [63], thereby increasing the power to detect QTLs. The Bonferroni method is considered the most conservative correction, and while it minimizes type I error (false positives), it does so by overly inflating type II error (false negatives) [64]. Consequently, small-effect QTLs are missed, especially in studies with small to moderate sample sizes and a high number of independent tests [65]. For this reason, here we highlighted those MTA that were significant even considering the most conservative threshold, but a suggestive significance threshold was also adopted to identify those markers that, even with low effect, contributed to the trait phenotypic variability. Additionally, considering the former LD results, the surrounding genomic region of every significant SNP was scanned to identify potential candidate genes based on their putative function, active transcription in the fruit, and bibliographic research (Appendix A). However, it must be noted that these genes are only an early proposal, based on the simultaneous fulfillment of three evidence criteria (positional, gene annotation, and expression in peach fruits), and further functional analyses are required to validate their function and their effects on phenotypic variation. Another limitation of this approach is that genes not correctly annotated in the genome assembly or with uncharacterized proteins may not be considered in this study. The most significant MTAs identified in the multi-year analysis, along with the fruit-expressed genes located within the genomic region with annotations matching the traits under study, are discussed below.

#### 3.4.1. Fruit Weight

The fruit weight is a trait known for its complex and polygenic control, with reported QTLs scattered throughout the peach genome [45,66]. Here, the strongest association signal (*p*-value = 5.52 × 10^−5^) was found on chromosome 6 (Pp06_3223902; Appendix A) for seasons 2021–2022, 2023–2024, and the multi-year dataset, and colocalized with the strong MTA detected for MD. Besides the candidate genes proposed for MD (see below), we considered *Prupe.6G050800*, which encodes an endo-1,3-beta-glucosidase 9 gene. According to the PeachMD database [32] its maximum expression in fruit occurs in S3, suggesting a role in cell wall remodeling and expansion. Similarly, association and linkage mapping in a non-flat peach population also identified that the start of chromosome 6 was highly associated with fruit size and shape [67], as well as two studies performing a GWAS: one in a population including landraces and modern breeding lines [52] and the other that; moreover, included wild relatives, wild peaches, and ornamental lines [68]. Similarly to our results, Cao et al. (2019) [68] did not detect this QTL in one of the three seasons evaluated, suggesting the environment may strongly influence it. Nevertheless, this coincidence in at least four populations with different backgrounds indicates that this region should be targeted for the molecular study of the trait.

We also identified an MTA with a candidate gene in proximity, located on chromosome 3 (Pp03_5361469; *p*-value = 0.0016), and another 8 MTAs near an OVATE domain-containing protein (*Prupe.3G069200*). Genes of this family are involved in plant development and growth [69,70], and accordingly, one *OVATE* gene in chromosome 6 was suggested as responsible for the flat peach phenotype [71]. In chromosome 8 we detected an MTA for the 2020–2021 (SNP Pp08_15930625; *p*-value = 4.34 × 10^−4^; PVE = 14.58%) containing genes annotated as Cyclin-dependent protein serine/threonine kinase inhibitor activity (*Prupe.8G122300*), ubiquitin protein ligase activity protein binding (*Prupe.8G140300*), and brassinosteroid signalling (*Prupe.8G140900*), all of them involved in processes related with cell replication in fruits [72,73,74]. Other genes found in the scanned MTAs region windows included xyloglucan endotransglucosylase/hydrolases, which are involved in cell wall restructuring, cell expansion, organ growth and fruit ripening [75]; pectin methylesterase, which are pivotal in the processes of plant growth and fruit ripening [76]; and an Auxin-binding protein T85 with a GO annotation suggesting a positive regulation of cell size.

#### 3.4.2. Solid Soluble Content

The highest association signal for SSC was on chromosome 5 (LG5_16786885; *p*-value = 3.75 × 10^−5^), which falls within the interval of previously reported QTLs for the trait [47,77]. Because the QTL was identified here using a GWAS approach in a diverse panel of individuals, and in Hernández-Mora et al. (2017) [47] by multi-family linkage mapping, marker-assisted selection of the trait may apply to other populations. Still, it is uncertain whether the QTL is stable across seasons, because even though it was detected here in the multi-year analysis and in the 2020–2021 season and in the three evaluated seasons in other studies [77], the signal was not detected in the 2021–2022 and 2022–2023 single-year analysis. Nevertheless, the marker explained 16% of the phenotypic variation in the multi-year analysis, which highlights the feasibility of marker-assisted selection for the trait. Still, no clear GWAS signal was detected for the trait in a diverse population consisting of wild peaches, landraces and improved cultivars [68], which might indicate that this QTL drives the variation existing only in modern cultivars. In this region, we have identified *Prupe.5G208500*, a fruit-expressed gene encoding an Agamous-like MADS-box protein AGL8 homolog (CMB1-like), related to FUL transcription factors. This protein is reported to regulate the expression of sugar transporters and partitioners in an ethylene-independent manner [78,79].

Still, as with the fruit weight, many other QTLs have been reported for the trait [11,77,80,81]. Here, the higher number of MTAs occurred in chromosome 4, with one of them (Pp04_16076936, *p*-value = 0.00336) located near a sorbitol dehydrogenase gene. Additionally, 29 MTAs were identified between Pp04:6638346-7543770, and a single SNP was found at position 2593706. Another relevant QTL was found in chromosome 1, with a single MTA (Pp01_12736773) in the proximity of a sucrose phosphate synthase (*Prupe.1G159700*), which in Chen et al. (2024) [82] was named *PpSPS2* and its expression correlated with sucrose accumulation in peach fruits.

#### 3.4.3. Maturity Date

A large-effect QTL on chromosome 4 has been identified for the MD in several *Prunus* crops, including peach, Japanese plum, apricot, and sweet cherry [83,84,85,86], suggesting a conserved mechanism within the genus. Two *NAC* transcription factors have been proposed as candidate genes, and functional analysis has demonstrated that they interact with each other to activate the transcription of genes related to ethylene biosynthesis, fruit softening, sugar accumulation, organic acid degradation, and cell elongation [87]. Accordingly, the highest GWAS signal in our study was observed for the MD and was located near these genes (Pp04_11112825, *p*-value = 2.53 × 10^−6^; PVE = 21%), confirming the high effect of the mutations previously reported for these *NAC* genes. The same signal has also been detected in peach GWAS experiments, further confirming the effect of this QTL in the MD [52].

Another strong GWAS signal was identified on chromosome 6 (Pp06_3311491; *p*-value = 4.97 × 10^−5^; PVE = 15%) in all the seasons included in this study. QTLS have been detected in this region in other studies involving stone fruits [85,88]. In the LD region surrounding this MTA, and according to the PeachMD database, we identified three expressed genes in fruits, which are annotated as being involved in the ripening process. *Prupe.6G039700* encodes for a ERF/PTI6-like, previously reported as activating ethylene biosynthesis genes *PpACS1* and *PpACO1* [89]; *Prupe.6G046900* encodes a WRKY33 protein, reported to be interacting with ERF7 transcription factor to activate ripening-associated processes such as color changes in tomato [90]; and *Prupe.6G042000* could be suggested as an effector of this process, since it encodes an Expansin A8, known to be an ethylene-dependent protein with a central role in cell-wall disassembly in ripening [91].

Other identified signals were close to polygalacturonase and pectate lyase genes, which play an active role in the breakdown of pectin in cell walls [92,93] and have been previously associated with fruit firmness [94]).

#### 3.4.4. IAD

The analysis of the index of absorption difference in the chlorophylls in fruit (IAD) revealed a strong association signal below the Bonferroni threshold (Pp02_19688605; *p*-value = 6.83 × 10^−6^; PVE = 19%), that was found in seasons 2020–2021, 2023–2024 and the multiyear dataset. In season 2021–2022, although in a different but near SNP (Pp02_19692053), we found an MTA significantly associated with this trait (4.42 × 10^−6^; PVE = 19.23%), confirming a certain stability of this QTL. Within the LD region surrounding Pp02_19688605, Prupe.2G135300 was the only candidate gene that was simultaneously (i) expressed in fruit, (ii) annotated as encoding for a chloroplast-localized protein, and (iii) putatively involved in ripening-related chlorophyll degradation. This gene encodes a plastid-lipid-associated FIBRILLIN/PAP protein that localizes to expanding and remodeling plastoglobules/lipoprotein particles as chloroplasts convert into chromoplasts during fruit ripening. In tomato fruit, plastoglobules recruit carotenoid biosynthetic enzymes, facilitating carotenoid accumulation while chlorophyll is degraded [95]. Moreover, genetic evidence indicates that the loss of fibrillin function (fbn2 CRISPR/Cas9-knockout Arabidopsis mutants) leads to accelerated chlorophyll degradation and faster senescence [96]. Together, these findings suggest this peach FIBRILLIN/PAP11 ortholog is involved in chromoplast biogenesis and pigment sequestration as the fruit loses its chlorophyll. Although this MTA has a strong and stable effect, the suggested interpretation of its possible mechanism in trait variability must be taken with care.

Another significant signal was identified in chromosome 4 for the multi-year dataset, specifically SNP Pp04_4119665 (*p*-value = 2.44 × 10^−4^; PVE = 12.82%). In the 2020–2021 and 2021–2022 seasons, MTAs were detected near this SNP, although with minor significance. In this region, we observed in the PeachMD database that *Prupe.4G082000* is expressed in fruits and encodes a chloroplastic 9-cis-epoxycarotenoid dioxygenase (NCED2), a rate-limiting enzyme in ABA biosynthesis in peach fruit [97], a phytohormone that, when exogenously applied, promotes fruit chlorophyll loss (degreening) alongside carotenoid accumulation in tomato fruit [98].

In addition, two other genes related to chlorophyll metabolism were identified near three MTAs, one of which was a pheophorbide a oxygenase (*Prupe.6G113600*). Recent transcriptomic analysis aimed at identifying genes related to chlorophyll degradation in peach skin has highlighted the importance of this specific gene (*PpPAO*), which was able to reduce chlorophyll levels in transiently transformed *Nicotiana benthamiana* leaves [99]. Because chlorophyll degradation is directly linked to fruit senescence, the markers identified here provide testable hypotheses about the genetic control of peel degreening and ripening behavior. From a breeding perspective, these *loci* could contribute to early enrichment of breeding populations with genotypes that mature earlier or later and exhibit more desirable ripening dynamics. However, given that IAD is a non-destructive, easy-to-phenotype trait and is strongly influenced by environment and management, these markers are more likely to be useful as components of multivariate genomic selection models or for early-stage population enrichment, rather than as stand-alone diagnostic tools for routine marker-assisted selection.

Although these findings do not demonstrate the genetic determinants of the studied traits, they offer a preliminary approach to dissect the potential mechanisms underlying their architecture, based on MTA signals obtained across the three phenotyping seasons and a multiyear BLUEs estimation.

## 4. Materials and Methods

### 4.1. Plant Material and Genotyping

The study’s plant material consisted of a collection of 140 peach trees, comprising 59 commercial cultivars originating from the United States, Italy, and Spain, as well as 81 genetically advanced lines from the Stone Breeding program at the University of Chile. All trees were planted in Rinconada de Maipú (Santiago, Chile) and maintained following standard agricultural practices.

Young leaves were collected from each tree and stored at −80 °C. Their DNA was extracted according to a modified CTAB protocol [100]. The DNA samples were sent to IGA Technology Services (Udine, Italy) for library preparation and sequencing. The library was constructed following the ddRADseq protocol [101] using *NspI* (frequent cutter) and *MboI* (rare cutter) (New England BioLabs, Ipswich, MA, USA) restriction enzyme combination, with 450–600 bp range fragment selection step in a BluePippin (Sage Science, Beverly, MA, USA) instrument and sequenced in a NovaSeq 6000 instrument (Thermo Fisher Scientific, Waltham, MA, USA) to generate 150 bp paired-end reads. The reads were demultiplexed using Stacks v2.61 [102], and the sequencing quality was assessed using FastQC v.11.9 (https://www.bioinformatics.babraham.ac.uk/projects/fastqc). All 140 samples passed the mean and per-position sequence quality thresholds, with an average of 4.3 million high-quality reads per sample. The reads were aligned to the reference genome of *Prunus persica* Whole Genome Assembly v2.0 [103] using BWA-MEM2 v.2.2.1 software [104]. The SNP calling was performed with Stacks v2.64 with the maruki-high model, limited to properly paired-end reads with a minimum mapping quality of 20, and filtering out SNPs with a genotype quality below 20. A stringent filter was applied with VCFtools [105] to keep only biallelic SNPs with a minor allele frequency (MAF) above 10%, a missing data rate of less than 3%, and in Hardy–Weinberg equilibrium (*p* > 0.05). To avoid SNP redundancy, the resulting 11939 SNPs were pruned based on linkage disequilibrium using Plink [106], removing one SNP for each pair of SNPs within a 10 kb window when their value exceeded r^2^ > 0.5. Sequencing, mapping and genotyping details for each sample are available in Appendix A. The chromosomal distribution of the resulting filtered markers was plotted in R 4.4.1 [107] using the CMplot package [108].

### 4.2. Evaluation of Phenology and Fruit Quality Traits

During the 2020–2021, 2021–2022 and 2023–2024 harvest seasons, 3 to 10 fruits per tree were harvested at the maturity stage. This was determined when most fruits on the tree had a pressure between 10 and 12 lb, measured with a manual fruit penetrometer FT 327 (Turoni, Forlì, Italy). The phenological and fruit quality traits evaluated were fruit weight (FW), soluble solid content (SSC), maturity date (MD) and chlorophyll absorbance index (IAD). The FW was determined with a digital scale, the SSC was assessed with a refractometer (ATAGO^®^ hand-held, %), and the IAD was quantified using a DA-meter (TR Turoni, Forli, Italy). The MD was scored as Julian days on the day of harvest. The distribution of the traits was visualized, and their modality was estimated by using the Modes function included in the LaplacesDemon R library. Their pair-wise correlation was calculated with Pearson tests.

An adjustment of the phenotypic records was performed prior to conducting the GWAS (see below). Best Linear Unbiased Estimates (BLUEs) were calculated for each genotype and trait using the following linear mixed model [109]:y = X_1_*b* + X_2_*g* + Z_1_*s* + *e*(1)y = X_1_*b* + X_1_*g* + *e*(2)
where *y* corresponds to phenotypic records. X_1_, X_2_ and Z_1_ are the incidence matrices that relate the vectors *b*, *g* and *s* to *y*. The *b* vector denotes the fixed effect of the overall mean. *g* and *s* are the vectors of fixed and random effects for genotype (BLUEs) and season, respectively. *e* is the vector for residual effects. The models 1 and 2 were used to analyze traits measured across the three consecutive seasons and independently within each season, respectively. The standard broad-sense heritability (H^2^) of each trait was calculated with formula (1) by using the H2cal function from inti R library.

### 4.3. Linkage Disequilibrium Pattern and Population Structure

Linkage disequilibrium (LD) pattern was assessed in TASSEL 5 software [110]. Default parameters were applied, and the LD critical value was determined according to Breshegello and Sorrells (2006) [31]. The LD curve was adjusted according to the method of Hill and Weir (1994) [111].

Population structure analysis was performed with no prior population assumption using Structure v.2.3.4 [112]. An admixture ancestry model with independent allele frequencies between populations was assumed, and a burn-in period and a total Markov chain Monte Carlo of 10,000 and 200,000 iterations, respectively, were used to estimate the Fst values and the Q matrix. The optimal number of genetic clusters (*K*) was selected according to Evanno et al., (2005) [57] by testing K = 1 to 10 in the Structure Harvester software [113]. In addition, a principal component analysis (PCA) with 5 putative components was performed in TASSEL 5. A maximum-likelihood (ML) tree was constructed using RAxML-ng v. 2.0 [114] with an all-in-one analysis including ML tree search and non-parametric bootstrap with the model LG+G8+F, 10 randomized parsimony starting trees, and 200 bootstrap replicates. The calculation of kinship was performed using the centered IBS method, in the TASSEL 5 software.

### 4.4. Genome-Wide Association Studies (GWAS) and Annotations

An association analysis was performed for the studied traits using the BLUEs from Equations (1) and (2) separately, where the results were analyzed with the two statistical significance [35]: (i) A Bonferroni correction (*p* < 8.5 × 10^−6^) based on the number of independent tests (n = 5861 total SNPs; *p* < 0.05); (ii) a suggestive *p*-value < 0.00625 (*n* = 8 chromosomes; *p* < 0.05), considering 8 independent association tests.

A Mixed Linear Model (MLM) was performed for each trait and season via TASSEL, as follows [115]:*g* = X*b* + S*a* + Q*v* + Z*u*(3)
where *g* corresponds to adjusted phenotypes for each trait (and season) extracted from models (1) and (2). The term X*b* contains fixed effects (excluding SNP and population structure effects) of overall mean. The *a*, *v* and *u* correspond to the vectors for SNP effects (fixed), structure population effects (fixed; two principal components considered) and polygenic background effects (genomic relatedness; random), respectively. *e* is a vector of residual effects.

For all significant marker–trait associations, a search for candidate genes was conducted within ±475,000 bp genomic windows, determined by the extent of linkage disequilibrium. For each gene annotated within these regions, functional prediction of their encoded protein sequences were obtained using PANNZER2 [116] annotation tool. Genes with former bibliography or with annotations functionally related to the studied traits were proposed as suitable candidates only if they were differentially expressed in fruit, in a developmental stage at which each trait is reported to be defined, according to 221 transcriptomes of peach in different cultivars and fruit development stages, available in the PeachMD database [32].

## 5. Conclusions

In this study, marker–trait associations and candidate genes were investigated to better understand the genetic architecture of peach fruit maturity, chlorophyll content, soluble solid content, and weight determination. The GWAS used a multi-season approach across three harvest seasons and identified stable QTL regions that determine the phenotypic variation in the studied traits. The study confirmed several previously reported marker–trait associations, including the relevance of the beginning of chromosome 6 to fruit weight, a QTL on chromosome 5 for soluble solid content, and chromosome 4 for maturity date in a region containing two *NAC* transcription factor genes. Additionally, novel putative regions have been proposed here; the most relevant is the IAD on chromosome 2, derived from a strong signal (Pp02_19688605) below the Bonferroni threshold and explaining 19% of the phenotypic variation. Suitable candidate genes have been proposed based on functional annotation and differential expression in peach fruit, although further functional experiments will be required to validate their function. The marker–trait associations reported here will be useful for marker-assisted breeding of the crop.

## Figures and Tables

**Figure 1 plants-15-00189-f001:**
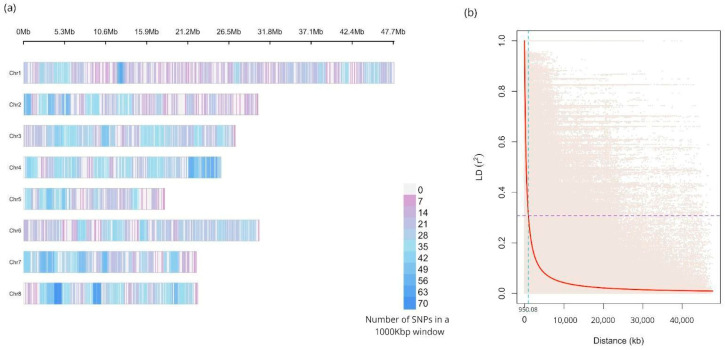
(**a**) Density plot of the 5861 single-nucleotide polymorphisms (SNPs) identified in the eight chromosomes of peach, represented by horizontal bars in Megabases (Mb). The color scale indicates the number of SNPs in a 1000 kb window. (**b**) Linkage disequilibrium (LD) decay pattern for the peach germplasm collection. The LD measures (*r*^2^) are plotted against physical distance between pairs of SNP markers. The purple line denotes the critical *r*^2^ value, while the turquoise line indicates the physical distance at which the critical value is reached.

**Figure 2 plants-15-00189-f002:**
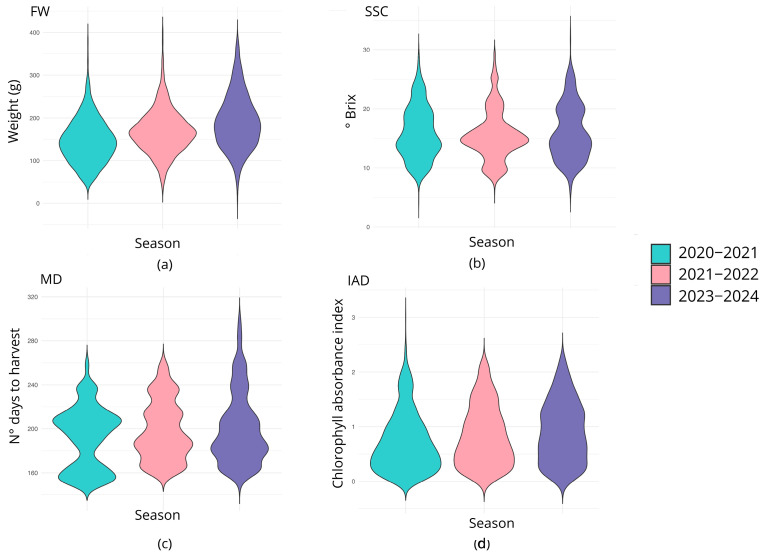
The distribution of values reflected per trait on an annual basis. In (**a**) the weight of the fruits is expressed in grams for the three seasons evaluated; (**b**) the solid soluble content is represented in ºBrix; (**c**) the maturity date is indicated in Julian days before harvest for the southern hemisphere, and (**d**), the IAD does not have a specific unit of measure given that the value is a ratio.

**Figure 3 plants-15-00189-f003:**
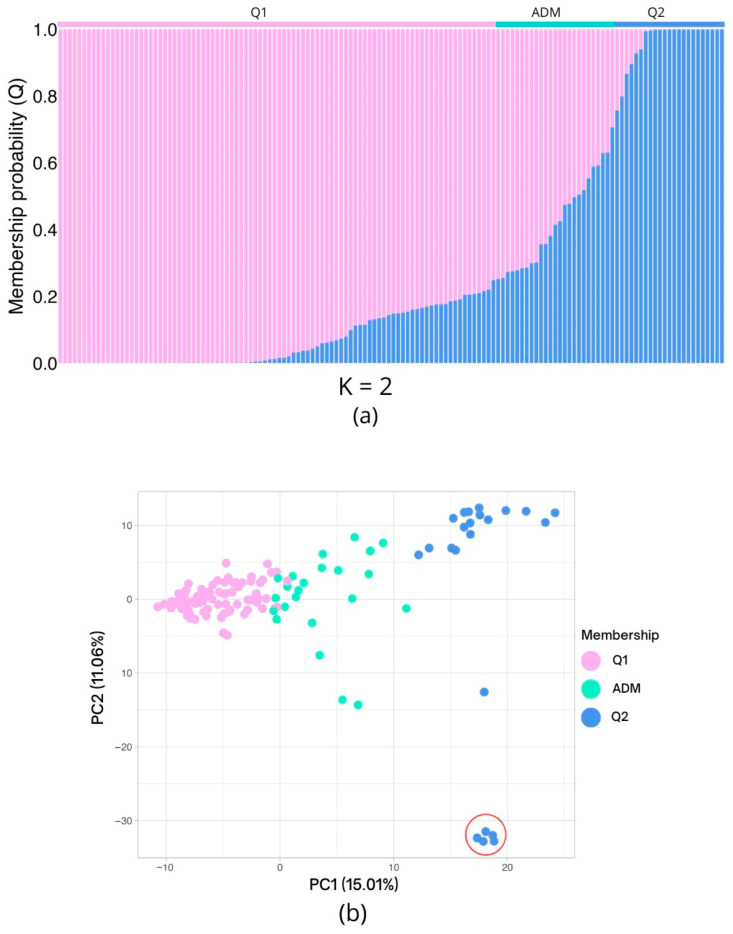
The population structure and principal component analysis (PCA) results are based on 5861 high-quality single-nucleotide polymorphisms (SNPs). (**a**) The inferred population structure of the 140 individuals indicates the existence of two genetic clusters (K = 2). The x-axis represents the individuals arranged according to their membership probability (y-axis; Q), with Q1 and Q2 representing the two distinct clusters. The bar at the upper limit denotes the allocation to a specific cluster, using a Q > 0.75 threshold. When this threshold is not attained, the individual is designated as admixed (ADM). (**b**) Genetic PCA plot, distributed in the coordinates generated on the first two components. The individuals are colored according to their former assignment. The red circle indicates five individuals from Q2 which are isolated from the remaining individuals of the same genetic cluster due to the genetic variation represented in PC2.

**Figure 4 plants-15-00189-f004:**
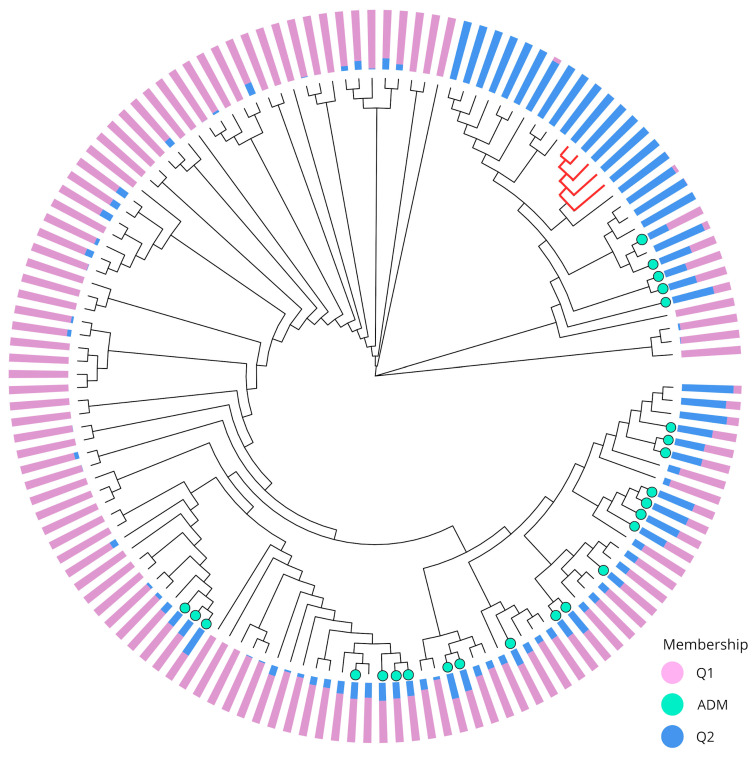
A Maximum Likelihood phylogenetic tree illustrating the genetic relationships and population structure among 140 individuals included in this study. The tree was constructed using 5861 SNPs. The colored bars at the tips of the branches indicate the inferred population membership for each individual, with Q1 (purple) and Q2 (blue) representing distinct genetic clusters. The light green colored tips represent the admixed individuals (ADM). The red branches correspond to five individuals assigned to Q2 but are isolated from most individuals of the same genetic group in the PCA.

**Figure 5 plants-15-00189-f005:**
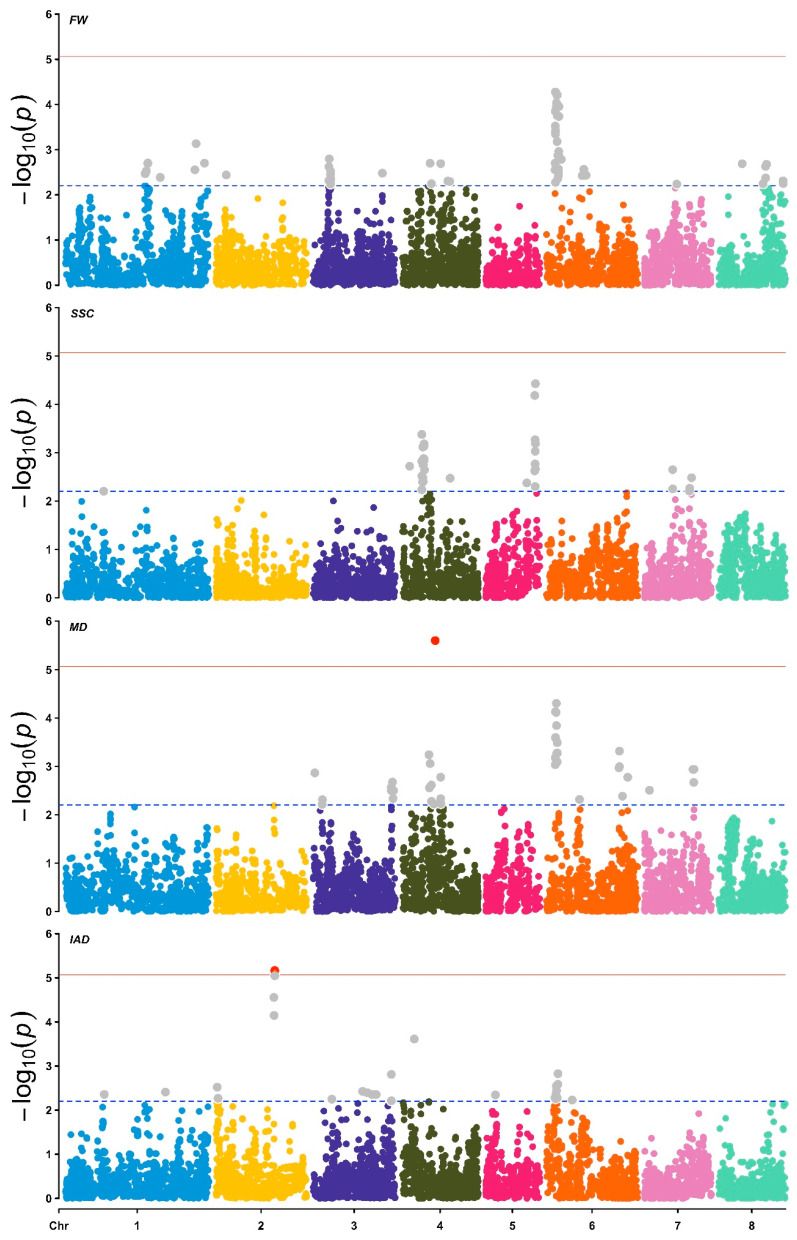
Manhattan plots of marker–trait associations (MTAs) for the studied traits, based on a multi-season approach. From the top to the bottom: Fruit weight (FW), soluble solid content (SSC), maturity date (MD), and the chlorophyll index (IAD). The dashed and solid horizontal lines represent the suggestive (*p*-value < 0.00625) and Bonferroni (*p*-value < 8.5 × 10^−6^) thresholds. Grey points represent MTAs over the suggestive *p*-value threshold, red points indicate MTAs over the Bonferroni threshold.

**Table 1 plants-15-00189-t001:** Summary of SNP markers and linkage disequilibrium distribution across the eight chromosomes of *Prunus persica*.

Chromosome	Chr. Size (Mbp)	N° Markers	Marker Density	N° Cross-Paired Markers	*r* ^2^
Chr1	47.86	956	2.00	456,490	0.1245
Chr2	30.43	695	2.28	241,165	0.1512
Chr3	27.39	769	2.81	295,296	0.1222
Chr4	25.86	859	3.32	368,511	0.1527
Chr5	18.50	452	2.44	101,926	0.2573
Chr6	30.79	715	2.32	255,255	0.1265
Chr7	22.39	683	3.05	232,903	0.1164
Chr8	22.58	732	3.24	267,546	0.1293

Chr. Size (Mbp): Chromosome size in millions of base pairs; N° Markers: marker count; Marker density: mean number of markers in 100 kilobase pairs. N° Cross-paired Markers: markers compared with other markers in each chromosome; *r*^2^: Average of linkage disequilibrium values.

**Table 2 plants-15-00189-t002:** Summary of results from the SNP-based association analysis (*p* < 0.00625) in the peach collection evaluated in three seasons and the multi-season analysis, for which candidate genes were found in the suggested genomic regions. The marker column indicates the marker name associated with each trait; this name is composed of the chromosome name and the position according to the *P. persica* genome V2.0 (i.e., marker Pp01_26499374 is in chromosome 1 at the position 26499374 bp). The window column indicates the position of the 950 kb genomic segment, considering the determination by the LD critical value (Section 2.2). To avoid redundancy, significant MTAs for two or more traits near a candidate gene (Appendix A) highlight in bold. Asterisk indicates a marker colocalizing QTLs for MD and FW, and included annotations considering both traits.

Trait	Marker	Window	*p*-Value	Gene	Protein Description	Annotation
FW	Pp01_26499374	26024374:26974374	0.00339	*Prupe.1G255100*	Xyloglucan endotransglucosylase/ hydrolase	xyloglucan:xyloglucosyl transferase activity; xyloglucan metabolic process, multidimensional cell growth, plant-type cell wall organization or biogenesis
	Pp01_26611800	26136800:27086800	0.00339			
	Pp01_26952521	26477521:27427521	0.00297			
	Pp01_31541184	31066184:32016184	0.00412	*Prupe.1G337000*	Xyloglucan endotransglucosylase/hydrolase	xyloglucan:xyloglucosyl transferase activity, xyloglucan metabolic process
	Pp01_31546083	31071083:32021083	0.00412			
	Pp01_31568771	31093771:32043771	0.00412			
	Pp01_31601793	31126793:32076793	0.00410	*Prupe.1G417900*	Pectin methylesterase CGR3	pectin metabolic process; methyltransferase activity
	Pp03_5240083	4765083:5715083	0.00482	*Prupe.3G069200*	OVATE domain-containing protein	Helicase activity
	Pp03_5246023	4771023:5721023	0.00239			
	Pp03_5361469	4886469:5836469	0.00160			
	Pp06_3223902	2748902:3698902	5.52 × 10^−5^	*Prupe.6G050800*	endo-1,3-beta-glucosidase 9	glucan endo-1,3-beta-D-glucosidase activity
	Pp08_8024146	7549146:8499146	0.00205	*Prupe.8G152500*	Xyloglucan endotransglucosylase/hydrolase	xyloglucan:xyloglucosyl transferase activity; xyloglucan metabolic process
	Pp08_15930625	15455625:16405625	4.34 × 10^−4^	*Prupe.8G122300;* *Prupe.8G140300;* *Prupe.8G140900*	F-box (SKIP22/FBXO-like, SCF E3 ligase); RING-type E3 ubiquitin (RHY1A-like); BAK1-like (BRI1 LRR-RLK co-receptor)	Cyclin-dependent protein serine/threonine kinase inhibitor activity; ubiquitin protein ligase activity; protein binding, brassinosteroid signalling
	Pp08_21623883	21148883:22098883	0.00491	*Prupe.8G247700*	Auxin-binding protein T85	auxin binding; positive regulation of cell size
	Pp08_21685700	21210700:22160700	0.00491			
	Pp08_21717973	21242973:22192973	0.00563			
	Pp08_21725549	21250549:22200549	0.00513			
SSC	Pp01_12736773	12261773:13211773	0.00625	*Prupe.1G159700*	Sucrose-phosphate synthase	sucrose-phosphate synthase activity; sucrose biosynthetic process
	Pp04_16076936	15601936:16551936	0.00336	*Prupe.4G240300*	Sorbitol dehydrogenase	oxidoreductase activity, acting on the CH-OH group of donors, NAD or NADP as acceptor
	Pp05_16778038	16328038:17228038	3.75 × 10^−5^	*Prupe.5G208500*	Agamous-like MADS-box protein AGL8 homolog (CMB1-like)	RNA polymerase II transcription regulatory region sequence-specific DNA binding, protein dimerization activity, 2-alkenal reductase [NAD(P)+] activity, positive regulation of transcription by RNA polymerase II
MD	Pp03_26053887	25578887:26528887	0.00316	*Prupe.3G287200*	Polygalacturonase	polygalacturonase activity
	Pp03_26061543	25586543:26536543	0.00270			
	Pp03_26442013	25967013:26917013	0.00211	*Prupe.3G296600*	Pectate lyase	pectate lyase activity; pectin catabolic process
	Pp03_26631520	26156520:27106520	0.00463			
	Pp03_26643124	26168124:27118124	0.00460			
	Pp03_26785134	26310134:27260134	0.00316	*Prupe.3G309400*	Omega-hydroxypalmitate O-feruloyl transferase	fruit ripening, climacteric; response to ethylene
	Pp03_2995684	2520684:3470684	0.00608	*Prupe.3G043600*	Putative polygalacturonase (Fragment)	polygalacturonase activity
	Pp03_3093068	2618068:3568068	0.00488			
	Pp04_9841821	9366821:10316821	0.00243	*Prupe.4G165100*	Pectate lyase superfamily protein domain-containing protein	polygalacturonase activity
	**Pp04_9871961**	9396961:10346961	0.00528			
	Pp04_11112825	10662825:11562825	2.53 × 10^−6^	*Prupe.4G186800, Prupe.4G187100*	NAC domain-containing protein 72, NAC transcription factor 25	Sequence-specific DNA binding, regulation of DNA-templated transcription
	**Pp06_2988972** *	2538972:3438972	1.97 × 10^−5^	*Prupe.6G039700, Prupe.6G046900, Prupe.6G042000*	ERF/PTI6-like; WRKY33-like; Expansin-A8 (ExpA8)	Regulation of DNA-templated transcription, ethylene-activated signaling pathway and DNA-binding transcription factor activity; Regulation of DNA-templated transcription and DNA-binding transcription factor activity; plant-type cell wall organization and anatomical structure morphogenesis
	Pp06_24178736	23703736:24653736	0.00107	*Prupe.6G242400*	Ethylene-responsive transcription factor RAP2-7	DNA-binding transcription factor activity
	Pp06_24240967	23765967:24715967	0.00104			
	Pp06_24315671	23840671:24790671	0.00101			
	Pp06_24318377	23843377:24793377	0.00048	*Prupe.6G247100*	Pectate lyase	pectate lyase activity; pectin catabolic process
	Pp06_25309512	24834512:25784512	0.00413			
IAD	Pp01_33303400	32828400:33778400	0.00389	*Prupe.1G353100*	Pheophytinase	chlorophyllase activity
	**Pp02_19688605**	19688605:20138605	6.83 × 10^−6^	*Prupe.2G135300*	Plastid-lipid associated protein PAP/fibrillin family protein	Chromoplast remodelling, plastid localization
	Pp04_4119665	3644665:4594665	2.44 × 10^−4^	*Prupe.4G082000*	9-cis-epoxycarotenoid dioxygenase NCED2, chloroplastic	carotene catabolic process; abscisic acid biosynthetic process; chloroplast stroma; carotenoid dioxygenase activity; metal ion binding
	Pp06_8576293	8101293:9051293	0.00589	*Prupe.6G113600*	Pheophorbide a oxygenase, chloroplastic	pheophorbide a oxygenase activity; chlorophyllide a oxygenase activity
	Pp06_8587833	8112833:9062833	0.00589			

## Data Availability

The genotypic (ddRAD-derived SNP) and phenotypic datasets that support the findings of this study will be made available upon reasonable request to the corresponding author after acceptance of the manuscript for publication. Requests should be addressed to the corresponding author listed on the title page. Any additional materials (e.g., scripts used for data processing and analysis) will likewise be shared upon request after acceptance, subject to reasonable data-sharing and ethical considerations.

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
