# Peer review of "Multi-Season Genome-Wide Association Study Reveals Loci and Candidate Genes for Fruit Quality and Maturity Traits in Peach"

_plants, 2026, doi:10.3390/plants15020189_

Round 1
Reviewer 1 Report
Comments and Suggestions for Authors
The aim of the study is to detect genetic associations and candidate genes controlling fruit quality and phenological traits in a panel of 140 peach trees.
The work is well structured.
Materials and Methods are appropriate and consistent with the aims.
Results and conclusions are congruent.
However, in order to improve the quality of manuscript, I suggest the following revisions:
- The “Abstract”
- is too long, please summarize it and focus on the main results achieved.
- In the ‘Introduction section’,
- Please cite some other interesting works, especially associated with omics studies, such as genomic and GWAS studies.
- In the ‘Results’ section,
- Please make the figures more readable, perhaps by separating them into multiple figures (for instance Figure 3)
- In the “Conclusion” section,
- I suggest stressing the impact resulting from the results achieved, highlighting the differences compared to the results obtained in previous studies.
- A careful rereading of the entire text is suggested. Please pay attention to the scientific names of plants which should be written in italics (Line 39), as well as the names of genes. There are also some oversights.
Some minor issues:
- Please check the references format, https://www.mdpi.com/journal/plants/instructions
Author Response
We sincerely thank the reviewer for their thorough and thoughtful review. We will address each of his/her comments point by point in the following section. We have highlighted our responses to each reviewer's comment in blue to facilitate reference and clarity:
Reviewer´s comment: The “Abstract” is too long, please summarize it and focus on the main results achieved.
Response: To improve the clarity of the Abstract section, we have considered the reviewer’s comment and reduced it to 197 words, in line with the journal's instructions for authors. Additionally, we have emphasized the main results obtained and their implications for marker-assisted crop breeding.
Reviewer´s comment: In the ‘Introduction section’, please cite some other interesting works, especially associated with omics studies, such as genomic and GWAS studies.
Response: We have considered the reviewer’s comment and have included some references relevant to the field of association studies for candidate genes in fruit species (Lines 59-69).
Reviewer´s comment: In the ‘Results’ section, please make the figures more readable, perhaps by separating them into multiple figures (for instance, Figure 3).
Response: We have reformatted Figures 3 and 5 to enhance their clarity and readability. We changed the layout of Figure 3 from horizontal to vertical to improve readability. Additionally, we changed the Manhattan plot layout from a circular to four horizontally elongated plots.
Reviewer´s comment: In the “Conclusion” section, I suggest stressing the impact resulting from the results achieved, highlighting the differences compared to the results obtained in previous studies.
Response: We have revised the Conclusion section to emphasize the impact of our study, noting that it confirms previously reported QTLs and reports new ones for the first time, along with their implications for peach marker-assisted breeding (Lines 735-744).
Reviewer´s comment: A careful rereading of the entire text is suggested. Please pay attention to the scientific names of plants which should be written in italics (Line 39), as well as the names of genes. There are also some oversights.
Response: We have thoroughly revised the entire manuscript to ensure that species names, gene names, and other relevant terms are formatted correctly, as suggested by the reviewer.
Reviewer´s comment: Please check the references format, https://www.mdpi.com/journal/plants/instructions
Response: The format of all references included in the manuscript has been checked, and modifications have been made where necessary.
Reviewer 2 Report
Comments and Suggestions for Authors
The manuscript, titled "Genome-wide association study of fruit maturity, chlorophyll content, soluble solids content, and fruit weight in peach germplasm resources across three growing seasons," conducted a GWAS analysis of multiple important agronomic traits in 140 peach accessions across three harvest seasons, revealing several relevant genetic markers and candidate genes. The study design was reasonable, the methodology comprehensive, and the content has practical breeding significance. However, several issues remain regarding scientific expression, data interpretation, and logical structure, requiring further revision and improvement.
Major comments
- The current title is too long. It is recommended to simplify it appropriately to enhance the article's appeal, for example: "Genome-wide association study of peach fruit maturity and quality traits".
- The article's topic is of practical significance, and the research results provide a theoretical basis for the molecular improvement of peach quality traits. However, the novelty of the research is not yet fully expressed. It is recommended that the authors further clarify the differences between this study and existing GWAS work on peaches, especially the uniqueness of the multi-year data integration and analysis.
- Regarding the issue of using BLUEs (best linear unbiased estimates) instead of BLUPs (best linear unbiased predictors) in phenotypic data analysis, authors are advised to provide a reasonable explanation, especially in cases where genotype-environment interactions exist.
- Regarding the Bonferroni correction threshold used in the association analysis, the authors should discuss more clearly its impact on statistical power with moderate sample sizes and assess potential false negatives (Type II error).
- Although the article proposed several candidate genes, some associations were only inferred based on positional relationships, and their functional relevance was weak. It is recommended that the authors further clarify the selection criteria for candidate genes (e.g., whether they are expressed in fruit, whether they have known biological functions, etc.).
- The explanation of the association between IAD and PAP/fibrillin protein is speculative. It is recommended that authors be more cautious in their statements and clearly explain the indirectness of the relevant evidence.
- The authors identified two genetic clusters and a mixture of individuals, and their analytical methods were reasonable. However, the impact of population structure on the GWAS results was not adequately discussed. It is recommended that the authors further explain how structural bias was controlled in the GWAS (e.g., whether to introduce a Q matrix or PCs into the model).
- Some traits (such as SSC) are described as multimodal, but no specific statistical tests are provided. It is recommended that the authors add relevant tests or reduce the assertion in the description.
- Regarding the heritability of traits, it is recommended to supplement the corresponding estimated values and conduct comparative analysis with relevant literature to support the interpretation of the results.
Minor issues
- Abbreviations (such as SSC, IAD, FW) should be defined when they first appear and used consistently throughout the text.
- Figure 5 (Ring Manhattan Map) contains a lot of information but has limited readability. It is recommended to simplify the diagram or provide magnified views of the main prominent areas.
- Citation numbers in the text (such as [30], [40], [76]) are missing from the reference list. It is recommended to check and complete them.
This research has significant application potential, but further improvements are needed in data interpretation and presentation. The authors are advised to carefully consider these suggestions and revise the entire paper accordingly to enhance its quality and academic value.
Comments on the Quality of English Language
The language of the text is generally standard, but some sentences are too long. It is recommended to break them down appropriately to improve readability.
Author Response
We sincerely thank the reviewer for their thorough and thoughtful review. We will address each of his/her comments point by point in the following section. We have highlighted our responses to each reviewer comment in blue to facilitate reference and clarity:
Reviewer´s comment: The current title is too long. It is recommended to simplify it appropriately to enhance the article's appeal, for example: "Genome-wide association study of peach fruit maturity and quality traits".
Response: We have considered the reviewer’s comment, and the new version of the manuscript is now entitled: Multi-Season Genome-Wide Association Study Reveals Loci and Candidate Genes for Fruit Quality and Maturity Traits in Peach.
Reviewer´s comment: The article's topic is of practical significance, and the research results provide a theoretical basis for the molecular improvement of peach quality traits. However, the novelty of the research is not yet fully expressed. It is recommended that the authors further clarify the differences between this study and existing GWAS work on peaches, especially the uniqueness of the multi-year data integration and analysis.
Response: We agree with the reviewer that the novelty of our study should be further emphasized. For this reason, we have highlighted the multi-year approach in the study title, as well as in the Abstract and Conclusion sections. We also indicated the novelty of the QTLs associated with IAD (Lines 616-625, Discussion section)
Reviewer´s comment: Regarding the issue of using BLUEs (best linear unbiased estimates) instead of BLUPs (best linear unbiased predictors) in phenotypic data analysis, authors are advised to provide a reasonable explanation, especially in cases where genotype-environment interactions exist.
Response: We appreciate the reviewer’s comment regarding the choice between using BLUEs and BLUPs in the phenotypic fitting analysis. In our study, we chose to use BLUEs because the genotypes evaluated corresponded to commercial cultivars and specific peach accessions, for which the individual performance of each genotype, rather than inference to a broader genetic population, was of primary interest. For this reason, we treated genotypes as fixed effects, aiming to obtain precise, directly interpretable estimates for each named variety rather than predictions of random genetic values. In addition, we had replications per tree and therefore required a unified phenotypic value per variety. The replication effect (arising from fruits collected randomly within each tree) was modeled as a random effect to capture experimental variability while keeping genotype effects fixed appropriately. Furthermore, we conducted a preliminary analysis to evaluate potential environmental influences (particularly the season effect), which was also modeled as a random effect. This analysis indicated that season did not have a significant impact on any of the phenotypes across years, reducing the need for a predictive framework such as BLUP, since no additional environmental variance needed to be absorbed into random genotypic effects.
Reviewer´s comment: Regarding the Bonferroni correction threshold used in the association analysis, the authors should discuss more clearly its impact on statistical power with moderate sample sizes and assess potential false negatives (Type II error).
Response: In response to the reviewer’s comment, we have added a brief discussion addressing the adoption of the Bonferroni correction, acknowledging that the reduction of the Type I error comes with the increase of Type II errors due to the inability to capture small-effect QTLs in a population with moderate sample size. We justify why the few marker-trait associations below this threshold are relevant, and why a suggestive threshold was also adopted in this study to identify associations missed in the former correction (Lines 474-480).
Reviewer´s comment: Although the article proposed several candidate genes, some associations were only inferred based on positional relationships, and their functional relevance was weak. It is recommended that the authors further clarify the selection criteria for candidate genes (e.g., whether they are expressed in fruit, whether they have known biological functions, etc.).
Response: We have considered the reviewer’s comments and have clarified the information regarding our criteria for defining potential candidate genes (Lines 737-741). In this regard, we only included the genes that both i) had annotated functions to the trait under study or existing bibliography and ii) were differentially expressed in the fruit according to 221 peach fruit transcriptomes from the PeachMD database.
Reviewer´s comment: The explanation of the association between IAD and PAP/fibrillin protein is speculative. It is recommended that authors be more cautious in their statements and clearly explain the indirectness of the relevant evidence.
Response: We acknowledge the limitations in Lines 600-614 (Discussion section) and 743-759 (Conclusion section) regarding the inability to identify genes not annotated in the genome, with unidentified functions (uncharacterized proteins), and a lack of functional experiments. We highlight that the genes identified here are only an early candidate gene proposal, and further work is required to validate them. As suggested by another Reviewer, we have reduced the Abstract length and removed any mention of the potential candidate genes identified in the study to avoid misunderstandings and keep it focused on proposals.
Reviewer´s comment: The authors identified two genetic clusters and a mixture of individuals, and their analytical methods were reasonable. However, the impact of population structure on the GWAS results was not adequately discussed. It is recommended that the authors further explain how structural bias was controlled in the GWAS (e.g., whether to introduce a Q matrix or PCs into the model).
Response: In light of the reviewer’s comments, we have detailed the definition of the population structure (Lines 696-701, Methodology). Additionally, we have provided a brief discussion on the potential impact of the structure component on the GWAS analysis (Lines 454-459).
Reviewer´s comment: Some traits (such as SSC) are described as multimodal, but no specific statistical tests are provided. It is recommended that the authors add relevant tests or reduce the assertion in the description.
Response: Rather than relying on subjective visual inspection of phenotypic distributions, we used the Modes function in the LaplacesDemon library to objectively assess their modality. The function returns the most probable modality for each of the tested distributions, estimated by examining changes in the probability density across the full range of values and counting the points where the distribution has a local peak (a modal). We have detailed the use of this specific function in line 670.
Reviewer´s comment: Regarding the heritability of traits, it is recommended to supplement the corresponding estimated values and conduct comparative analysis with relevant literature to support the interpretation of the results.
Response: In this revised version of the manuscript, we have included the broad-sense heritability values (H²) for each of the studied traits (Lines 177-179 in Results; Lines 683-684 in Methodology). Additionally, as suggested by the reviewer, we have used these values to enhance the discussion of our phenotypic results in Lines 410-416, including the comparison with recent literature, which also reports heritability calculations for the MD, FW, and SSC traits.
Reviewer´s comment: Abbreviations such as SSC, IAD, FW) should be defined when they first appear and used consistently throughout the text.
Response: The manuscript has been modified to ensure that trait abbreviations are presented adequately.
Reviewer´s comment: Figure 5 (Ring Manhattan Map) contains a lot of information but has limited readability. It is recommended to simplify the diagram or provide magnified views of the main prominent areas.
Response: To ensure a clear and comprehensive presentation of our findings, the GWAS results have been shown using traditional Manhattan plots (Figure 5) based on the multi-year analysis, which are included in the main manuscript.
Reviewer´s comment: Citation numbers in the text (such as [30], [40], [76]) are missing from the reference list. It is recommended to check and complete them.
Response: We have thoroughly reviewed all references throughout the entire manuscript.
Reviewer´s comment: This research has significant application potential, but further improvements are needed in data interpretation and presentation. The authors are advised to carefully consider these suggestions and revise the entire paper accordingly to enhance its quality and academic value.
Response: We thank the reviewer for recognizing the relevance and potential impact of our research on fruit crop genetic improvement. As a team, we have carefully addressed the reviewer’s comments to improve our manuscript.
Reviewer 3 Report
Comments and Suggestions for Authors I have reviewed the manuscript titled "Genome-Wide Association Study of Fruit Maturity, Chlorophyll Content, Solid Soluble Content and Weight in a Peach Germplasm Collection across Three Seasons." This study presents a comprehensive GWAS analysis of key agronomic traits in peach, leveraging a diverse germplasm panel and multi-season data. The research is well-designed and addresses an important topic for peach breeding. However, several aspects require revision to enhance clarity, methodological rigor, and interpretation.
1.The current title is descriptive but could be more concise. Consider rephrasing to: "Multi-Season Genome-Wide Association Study Reveals Loci and Candidate Genes for Fruit Quality and Maturity Traits in Peach (Prunus persica)." This emphasizes the multi-season approach and key findings.
2.The abstract should briefly highlight the novelty of the study (e.g., use of a diverse panel and multi-season analysis) and explicitly state the number of significant SNPs identified. Also, include the practical implications for marker-assisted breeding more clearly.
3.Provide more details on the ddRADseq protocol, such as the specific restriction enzymes used (NspI and Mbol are mentioned but not emphasized), sequencing depth per sample, and quality control metrics (e.g., average coverage). This enhances reproducibility.
4.The use of Bonferroni correction (p < 8.5e-6) is conservative. Justify why this threshold was chosen over false discovery rate (FDR) or other methods, especially given the polygenic nature of traits. Mention if any sensitivity analyses were conducted.
5.Figure 2 (phenotypic distributions) is critical but lacks clarity in the caption. Ensure the caption explicitly describes the units and transformations (e.g., "without data transformation" is vague). Consider adding subpanels for each trait-season combination for better readability.
6.The discussion should directly compare identified QTLs with those from prior peach GWAS (e.g., Cao et al. 2016, Li et al. 2019). Highlight consistencies (e.g., chromosome 4 MD QTL) and novel findings (e.g., IAD-associated genes).
7.Add a dedicated paragraph on study limitations, such as sample size (n=140), which may limit power for detecting small-effect QTLs, and the impact of environmental variability on trait measurements across seasons.
8.Ensure all figures and tables are cited in the text in sequential order. For example, Figure 3 and 4 are mentioned but not effectively integrated into the results narrative. Describe key insights from these figures (e.g., PCA and phylogenetic tree) in the results section.
Author Response
We sincerely thank the reviewer for their thorough and thoughtful review. We will address each of his/her comments point by point in the following section. We have highlighted our responses to each reviewer’s comment in blue to facilitate reference and clarity.
Reviewer´s comment: The current title is descriptive but could be more concise. Consider rephrasing to: "Multi-Season Genome-Wide Association Study Reveals Loci and Candidate Genes for Fruit Quality and Maturity Traits in Peach (Prunus persica)." This emphasizes the multi-season approach and key findings.
Response: We have considered the reviewer’s comment, and the revised manuscript is now titled: Multi-Season Genome-Wide Association Study Reveals Loci and Candidate Genes for Fruit Quality and Maturity Traits in Peach.
Reviewer´s comment: The abstract should briefly highlight the novelty of the study (e.g., use of a diverse panel and multi-season analysis) and explicitly state the number of significant SNPs identified. Also, include the practical implications for marker-assisted breeding more clearly.
Response: We have shortened the Abstract and addressed the Reviewer’s comment by including the total number of associations, as well as emphasizing the use of single- and multi-season analysis and the implications of this work for marker-assisted breeding in the crop.
Reviewer´s comment: Provide more details on the ddRADseq protocol, such as the specific restriction enzymes used (NspI and Mbol are mentioned but not emphasized), sequencing depth per sample, and quality control metrics (e.g., average coverage). This enhances reproducibility.
Response: We have added more details about the sequencing and genotyping results in Lines 97-101. Additionally, Supplementary Table S1 now includes information per sample—such as the number of reads, number of loci, mean read depth before and after variant filtering, and genotyping rate. We have also provided more details about the enzymes and sequencing protocol in the methodology section (Lines 640-642).
Reviewer´s comment: The use of Bonferroni correction (p < 8.5e-6) is conservative. Justify why this threshold was chosen over false discovery rate (FDR) or other methods, especially given the polygenic nature of traits. Mention if any sensitivity analyses were conducted.
Response: We acknowledge that the Bonferroni correction (p < 8.5×10⁻⁶) is conservative, especially for polygenic traits where many true associations may have small effects. Our goal was to reduce type I error and ensure that the reported associations indicate robust signals that remain significant even under strict multiple-testing correction. Because there are relatively few independent markers after marker quality filtering and LD pruning, the Bonferroni threshold provided a clear and reproducible criterion. We agree with the reviewer's point that FDR is less conservative and better suited to the highly polygenic architecture typical of complex traits, but our conclusions were based only on the most robust signals. We have discussed this in Lines 473-479.
Reviewer´s comment: Figure 2 (phenotypic distributions) is critical but lacks clarity in the caption. Ensure the caption explicitly describes the units and transformations (e.g., "without data transformation" is vague). Consider adding subpanels for each trait-season combination for better readability.
Response: In figure 2, we included subpanels to improve the readability of the plots.
Reviewer´s comment: The discussion should directly compare identified QTLs with those from prior peach GWAS (e.g., Cao et al. 2016, Li et al. 2019). Highlight consistencies (e.g., chromosome 4 MD QTL) and novel findings (e.g., IAD-associated genes).
Response: We have included extensive comparisons in the Discussion section, such as QTLs matching other studies that reinforce their significance in peach breeding, along with the novel findings related to IAD and an explanation of why other studies might not have incorporated this trait in their association analysis (Lines 505-510, 539-541, 564-566, 615-618).
Reviewer´s comment: Add a dedicated paragraph on study limitations, such as sample size (n=140), which may limit power for detecting small-effect QTLs, and the impact of environmental variability on trait measurements across seasons.
Response: Although we did not explicitly identify the sample size as a limitation in our study, we had previously discussed its relevance. Therefore, we believe it is important to highlight that this aspect, along with the difficulty of capturing low-effect QTLs, also presents a challenge for us. We have revised a specific paragraph to explain and contextualize its impact on our analyses (Lines 474-479) and to highlight a case where multi-season analysis outperformed single-season analysis (Lines 539-541).
Reviewer´s comment: Ensure all figures and tables are cited in the text in sequential order. For example, Figure 3 and 4 are mentioned but not effectively integrated into the results narrative. Describe key insights from these figures (e.g., PCA and phylogenetic tree) in the results section.
Response: We have considered the reviewer's comment and carefully verified that all figures and tables are referenced correctly in the text. However, please note that if the manuscript is accepted, its formatting might undergo further changes during the editorial process.
Round 2
Reviewer 2 Report
Comments and Suggestions for Authors
accept
Comments on the Quality of English LanguageThe language of the text is generally standard, but some sentences are too long. It is recommended to break them down appropriately to improve readability.
Reviewer 3 Report
Comments and Suggestions for Authors
This paper has met the requirements of the journal and is recommended for publication.